# Comparison of measured LDL cholesterol with calculated LDL-cholesterol using the Friedewald and Martin-Hopkins formulae in diabetic adults at Charlotte Maxeke Johannesburg Academic Hospital/NHLS Laboratory

**Mogomotsi Dintshi** [ORCID] *, **Ngalulawa Kone, Siyabonga Khoza** [ORCID]

Departement of Chemical Pathology, National Health Laboratory Services and University of Witwatersrand, Johannesburg, South Africa

* mogomotsi.dintshi@nhls.ac.za

## Abstract

### Background

The National Cholesterol Education Programme Adult Treatment Panel III (NCEP ATP III) and the European Society of Cardiology recommend using low-density lipoprotein cholesterol (LDL-C) as a treatment target for cholesterol lowering therapy. The Friedewald formula underestimates LDL-C in non-fasted and hypertriglyceridemia patients. This study aimed to compare measured LDL-C to calculated LDL-C in diabetic patients using the Friedewald and Martin-Hopkins formulae.

### Methods

The data of 1 247 adult diabetes patients were retrospectively evaluated, and included triglycerides (TG), LDL-C, total cholesterol, and high-density lipoprotein cholesterol that were measured on the Roche Cobas® c702. Passing-Bablok regression analysis was used to determine the degree of agreement between measured LDL-C and calculated LDL-C using both formulae. The Bland-Altman plots were used to assess the bias at medical decision limits based on the 2021 European Society of Cardiology (ESC) guidelines on cardiovascular disease prevention in clinical practice.

### Results

Both formulae showed a good linear relationship against measured LDL-C. However, the Martin-Hopkins formula outperformed the Friedewald formula at LDL-C treatment target <1.4mmol/L. The Friedewald formula and the Martin-Hopkins formula had 14.9% and 10.9% mean positive bias, respectively. At TG-C ≥1.7 mmol/L, the Martin-Hopkins formula had a lower mean positive bias of 4.2% (95% CI 3.0–5.5) compared to the Friedewald

**Data Availability Statement:** The relevant data are uploaded to protocols.io at doi.org/10.17504/protocols.io.yxmvm2ky6g3p/v1.

**Funding:** The author(s) received no specific funding for this work.

**Competing interests:** The authors declare that no competing interest exist.

formula, which had a mean positive bias of 21.8% (95% CI 19.9–23), which was higher than the NCEP ATP III recommended total allowable limit of 12%.

## Conclusion

The Martin-Hopkins formula performed better than the Friedewald formula at LDL-C of 1.4 mmol/L and showed the least positive bias in patients with hypertriglyceridemia.

## Introduction

Cardiovascular disease (CVD) accounts for nearly 40 million deaths per annum worldwide [1] of which diabetes mellitus is a major risk factor. According to the International Diabetes Federation, diabetes mellitus is estimated to affect 463 million adults worldwide and over 19 million adults in the African region in 2019 [2]. The number of people living with diabetes mellitus is predicted to rise to 700 million adults by 2045, which represents an alarming 51% increase [2,3]. Insulin resistance is a hallmark of diabetes mellitus, which is strongly linked to dyslipidemia [4]. The lipid profile pattern associated with diabetes mellitus is elevated triglycerides (TG), low high-density lipoprotein (HDL), and elevated small dense low-density lipoprotein (LDL) [5,6]. This is consistent with the findings of the Framingham study, which found that hypercholesterolemia is a risk factor for CVD and predicts CVD risk over a 10-year period [7].

LDL cholesterol (LDL-C) is recognized as a risk factor for CVD by the National Cholesterol Education Programme Adult Treatment Panel III (NCEP ATP III) as well as the European Society of Cardiology (ESC) and it is used as a treatment target for cholesterol-lowering therapy [8,9]. Patients are firstly categorized using the Framingham risk score [10] based on their risk factors and then each class of patients is assigned a target LDL-C concentration to which lipid-lowering medications are targeted [8]. This emphasizes the need to accurately report LDL-C levels to improve patient classification and management.

In the laboratory, LDL-C can be assessed using a variety of direct or indirect measurement methods. The gold standard method for LDL-C is the beta quantification by ultracentrifugation [11]. However, this procedure is not ideal for routine clinical laboratory usage because it is cumbersome and time-consuming, and it also requires expert skills and a large sample volume [12]. Various formulae for calculating LDL-C have been developed to solve these issues [13]. Friedewald et al. originally proposed a formula to calculate LDL-C [14] and devised a method based on three parameters which include total cholesterol, triglycerides, and high-density lipoprotein (HDL). However, this formula has limitations: it underestimates the LDL-C in non-fasted individuals with high chylomicron levels, and it is invalid at triglycerides (TG) of > 4.5 mmol/L [12] as it assumes a constant 5:1 ratio relationship between triglycerides and VLDL [15], even though the ratio has been proven to vary within and between individuals [16]. Moreover, it relies on the accurate measurement of total cholesterol, HDL-C, and TG parameters used in the calculation. As a result, serum samples from non-fasted patients who have conditions associated with high TG levels such as uncontrolled diabetes mellitus, dysbetalipoproteinemia or alcoholism, may underestimate calculated LDL-C [17]. This could lead to patients being misclassified for lipid-lowering treatment and predispose them to risk for cardiovascular complications.

Several studies have since revealed these flaws, and others have attempted to develop new formulae that better correlate with measured LDL-C testing. One such formula is the Martin-

Hopkins formula which is recommended by both the European Federation of Clinical Chemistry and Laboratory Medicine (EFLM) and the European Atherosclerosis Society (EAS) because of the advantage it offers over the Friedewald formula, particularly at LDL-C concentrations of <1.8 mmol/L, TG concentration 2.0–4.5 mmol/L and, in non-fasting samples [18]. Several studies have shown a good correlation between the Friedewald formula and the Martin-Hopkins formula in the general population [15,19], it is important that this formula is assessed in different populations before widely used. This study aims to compare measured LDL-C with calculated LDL-C using the Friedewald and Martin-Hopkins formulae in diabetic patients over a range of LDL-C and TG concentrations.

## Materials and methods

### Study design

We retrospectively reviewed the lipid profile data that was requested from Charlotte Maxeke Johannesburg Academic Hospital's (CMJAH) adult diabetic outpatient department (OPD) from August 2016 to December 2019. Before August 2016, LDL-C was determined using the Friedewald formula at CMJAH's National Health Laboratory Service (NHLS); but thereafter, an LDL homogeneous assay was adopted.

This study complied with the institutional regulations and was approved by the Human Research Ethics Committee (Medical) of the University of the Witwatersrand (clearance certificate No. M200858) and followed the Declaration of Helsinki. The data for the study was requested from the NHLS Central Data warehouse (CDW) after obtaining permission from Academic Affairs and Research Management System (AARMS), approval number PR20218. The data included triglycerides (TG), total cholesterol (TC), low-density lipoprotein cholesterol (LDL-C), total cholesterol (TC) as well as high-density lipoprotein cholesterol (HDL-C).

A total of 1 302 participants who met the inclusion criteria were enrolled in the study. The following inclusion criteria were used: adult participants of ≥18 years old, both male and female participants from the diabetic OPD whose full lipid profile results were performed at CMJAH's NHLS. Participants who were non-diabetics, <18 years old, and with incomplete lipid profile were excluded.

### Laboratory tests measurements

The Roche Cobas® c702 (Roche Diagnostics, Mannheim, Germany) was used to analyze lipid profiles according to manufacturer's instructions.

In summary, LDL-C was measured using a method that involves selectively solubilizing it with a surfactant which then combines with cholesterol esters and oxidase to yield $\Delta^4$-cholestenone + hydrogen peroxide ($H_2O_2$). The $H_2O_2$ reacts with peroxidase, and generates a red purple pigment [20]. For TC measurement, the first step consists of cholesterol esters cleavage and oxidation, with the resulting products triggering oxidative coupling of phenol and 4-aminophenazone to form a red quinone-imine dye that is quantified in the presence of peroxidase.

The measurement of HDL-C requires that LDL-C, VLDL-C, and chylomicrons be excluded, by complexing them with dextran sulfate in the presence of magnesium ions. HDL-C esters are cleaved and oxidized into $\Delta^4$-cholesterol and $H_2O_2$. The latter ($H_2O_2$), then reacts with $\Delta^4$-amino-antipyrine and sodium N-(2-hydroxy-3-sulfopropyl)-3,5-dimethoxyaniline in the presence of peroxidase to form a purple-blue color. Lastly, in the TG method, the free glycerol is first blanked from the reaction before the hydrolysis of triglycerides by lipase. The liberated glycerol undergoes multiple subsequent enzyme catabolism to finally form a colored compound, 4-(p-benzoquinone-monoimino)-phenazone. The intensity of the color of

various components of the lipid profile is directly proportional to the concentration of the analyte of interest.

We then estimated LDL-C from the given lipid profile results, using Friedewald and Martin-Hopkins formulae to assess the correlations between each formula and measured LDL-C, as well as between the two formulae.

The Friedewald formula used was as follows:

$$LDL - cholesterol\ (mmol/L) = Total\ cholesterol - HDL - \frac{Triglyceride}{2.2}$$

For Martin-Hopkins calculated LDL-C, the following formula was applied:

$$LDL - cholesterol\ (mmol/L) = Total\ cholsterol - HDL + \frac{Triglyceride}{adjustable\ factor}$$

The adjustable factor was obtained from 180-cell table strata using the participants' TG and non-HDL providing a more individualized approach. Non-HDL-C was obtained by subtracting HDL-C from total cholesterol. LDL-C in the Martin-Hopkins formula was calculated using a Microsoft Excel spreadsheet obtained from Johns Hopkins Medicine [21].

## Statistical analysis

The data were entered and categorized using Microsoft Office Excel 2016 (Microsoft, Seattle, WA, USA). Statistical analyses were performed using MedCalc for Windows, version 19.8 (MedCalc Software, Ostend, Belgium). The Tukey test was performed for outlier detection and a total number of 54 results were excluded from the 1301 original sample size. The D'Agostino-Pearson test was used for normal distribution testing. Normally distributed data were expressed as mean and standard deviation. The LDL-C data was categorized based on the 2021 European Society of Cardiology (ESC) guidelines on cardiovascular disease prevention in clinical practice [9]. The following LDL-C (mmol/L) treatment cut-offs were used; <1.40, <1.80, <2.60 and <3.00 [9]. As there are presently no published therapeutic treatment goals for triglycerides, we classified TG levels as low and high risks if they were less and more than 1.7 mmol/L, respectively.

The method comparison between measured and calculated LDL-C formulae was performed according to Passing-Bablok regression analysis. As stated earlier, the reference method, beta-quantitation by ultracentrifugation, was not fit for use in our busy routine setting as it is time-consuming and expensive. Hence, we used the Bland- Altman plots to determine and demonstrate the degree of agreement as well as the level of bias between measured LDL-C and calculated LDL-C. For reason stated above, the beta quantitation was not used as a reference method for our analysis.

The total allowable limits of ±12% which is recommended by NCEP ATP III [8], was used to measure the agreement between measured LDL-C and calculated LDL-C. A p-value of <0.05 was considered statistically significant.

## Results

The study comprises a total of 1247 participants, including 57.2% females and 41.5% males. The study populations' mean age was 55.4 years with a standard deviation (SD) of 15.5. The characteristics of the study population are shown in Table 1.

We first compared the calculated LDL-C derived from the Friedewald formula to the measured LDL-C. The regression analysis equation for Friedewald formula was y = -0.201 + 0.966x

**Table 1. Demographic and biochemical characteristics expressed as mean ± SD.**

| Variable | Mean ± SD |
|---|---|
| Age (years) | 55.3 ±15.5 |
| • Males | 53.4 ± 15.2 |
| • Females | 57.2 ± 15.4 |
| Total cholesterol (mmol/L) | 4.07 ±0.95 |
| • Males | 2.42 ± 0.81 |
| • Females | 2.44 ± 0.80 |
| • Unknown (N = 16) | 2.30 ± 0.60 |
| Triglyceride (mmol/L) | 1.45 ±1.02 |
| Non-HDL-C (mmol/L | 2.79 ±0.92 |
| LDL-C (measured) | 2.44 ±0.80 |
| LDL-C (Friedewald formula) | 2.13 ±0.78 |
| LDL-C (Martin-Hopkin Formula) | 2.22 ±0.78 |

LDL- C, low density lipoprotein cholesterol, HDL-C, high density lipoprotein cholesterol and SD, standard deviation.

(r = 0.952, 95% confidence interval (CI) 0.947 to 0.957, p-value <0.0001). (Fig 1A). A good linear relationship was observed between the methods.

We then compared the calculated LDL-C derived from the Martin-Hopkins formula to the measured LDL-C. The regression analysis equation for calculated LDL-C using Martin-Hopkins with measured LDL-C was y = -0.134 + 0.963 x (r = 0.954, 95% CI 0.949 to 0.959, p-value <0.0001), (Fig 1B). The Martin-Hopkins formula showed a better linear relationship with measured LDL-C when compared to the Friedewald formula.

Finally, when we compared the Martin-Hopkins formula to the Friedewald formula, we found that the regression analysis equation y = 0.0200 + 1.000 x had an even greater correlation. The correlation coefficient was 0.960, 95% CI 0.956 to 0.964 with a p-value p <0.0001, (Fig 1C).

To improve the comparison between the measured and calculated LDL-C, the regression analysis equations were obtained at different LDL-C treatment targets (Table 2 and S1 Fig). The Martin-Hopkins formula performed better than the Friedewald formula at LDL-C of 1.4 mmol/L. There was a positive linear relationship between calculated and measured LDL-C methods. At LDL-C of 2.6 mmol/L and ≥3.0 mmol/L, we found a significant correlation coefficient with the Martin-Hopkins formula when compared to the Friedewald formula (Table 2).

The Bland-Altman plots for Friedewald and Martin-Hopkin formulae showed a mean positive bias of 14% (95% CI 14.1–15.6) and 10.24% (95% CI 9.59–10.90), respectively (See Fig 2A and 2B).

Significant correlations were found across the measured and calculated LDL-C at TG of < 1.7 mmol/L and ≥ 1.7 mmol/L with Spearman correlation coefficient ranging from 0.962 to 0.949. There were 349 (28%) participants who had high-risk triglyceride concentrations (≥ 1.7mmol/L). At TG-C of ≥1.7 mmol/L, the Friedewald formula showed a mean positive bias of 21.8% (95% CI 19.9–23) which was higher than NCEP ATP III's recommended total allowable limits of 12%. At TG-C of ≥1.7 mmol/L, the Martin-Hopkins formula showed a reduced mean positive bias of 4.2% (95% CI 3.0–5.5) (Table 3 and Fig 3A and 3B).

## Discussion

We compared measured LDL-C to calculated LDL-C using the Friedewald and Martin-Hopkins equations in this study, which looked at the correlation of LDL-C methodologies in the

(A)

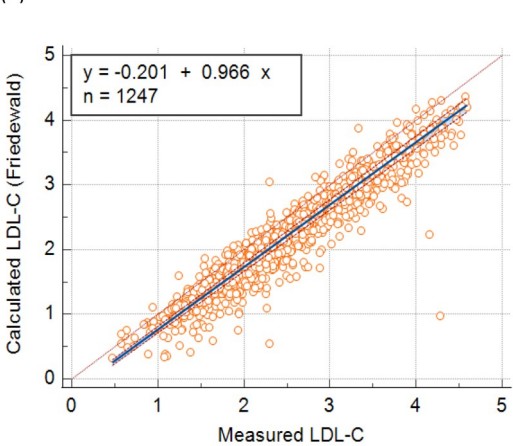

(B)

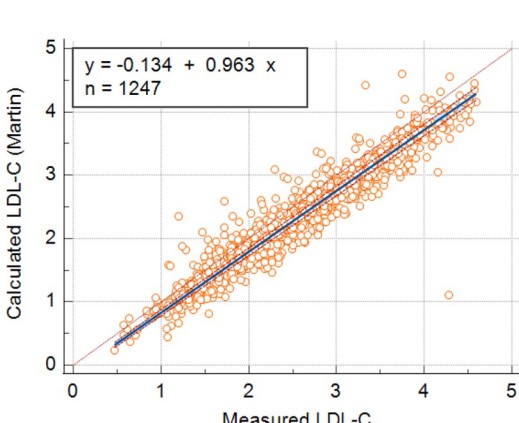

(C)

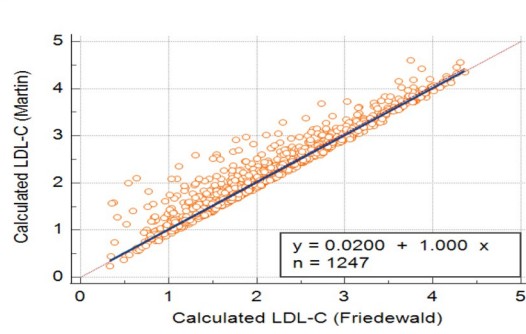

**Fig 1. The comparison of calculated to measured low-density lipoprotein (LDL-C) using Passing- Bablok regression analysis.** (A) Friedewald formula and measured LDL-C. (B) Martin Hopkins formula and measured LDL-C. (C) Friedewald formula and Martin Hopkins formula.

South African diabetic community. By using linear regression analysis, we were able to demonstrate that both formulae had a good correlation. However, the Martin-Hopkins formula proved to have a better correlation, particularly at lower LDL-C concentrations of <1.4 mmol/L and had the least mean positive bias when compared to the Friedewald formula. The Friedewald formula significantly underestimates LDL-C in hypertriglyceridemia.

**Table 2. Summary of the comparison between measured and calculated low-density lipoprotein (LDL-C), using the Friedewald and Martin Hopkin formulae as well as regression analysis at different LDL-C levels.**

| LDL-C level | N number | Correlation coefficient ($r^2$) | Gradient (95% confidence interval) | Y intersection (95% confidence interval) | p-value |
|---|---|---|---|---|---|
| <1.40 mmol/L | 95 | Friedewald 0.650 | 1.29 (1.00 to 1.70) | -0.50 (-01.00 to 0.19) | <0.0001 |
| | | Martin-Hopkins 0.682 | 1.43 (1.16 to 1.91) | -0.65 (-1.25 to −0.33) | <0.0001 |
| <1.80 mmol/L | 187 | Friedewald 0.576 | 2.33 (2.00 to 2.78) | -2.39 (-3.13 to −1.85) | <0.0001 |
| | | Martin-Hopkins 0.499 | 2.44 (2.00 to 3.00) | -2.49 (-3.37 to −1.77) | <0.0001 |
| <2.60 mmol/L | 482 | Friedewald 0.710 | 1.30 (1.20 to 1.40) | -0.92 (-1.15 to −0.72) | <0.0001 |
| | | Martin-Hopkins 0.745 | 1.26 (1.17 to 1.35) | -0.77 (-0.96 to −0.58) | <0.0001 |
| <3.00 mmol/L | 191 | Friedewald 0.539 | 2.29 (1.94 to 2.81) | -3.90 (-5.34 to −2.94) | <0.0001 |
| | | Martin-Hopkins 0.521 | 2.20 (1.83 to 2.67) | -3.58 (-4.90 to −2.56) | <0.0001 |
| ≥3.10 mmol/L | 292 | Friedewald 0.798 | 1.13 (1.06 to 1.22) | -0.85 (-1.15 to −0.57) | <0.0001 |
| | | Martin-Hopkins 0.831 | 1.11 (1.04 to 1.19) | -0.70 (-0.97 to −0.45) | <0.0001 |

These results build on existing evidence by Rossouw et al. in a study also performed in the South African population. This latter study evaluated approximately 14000 out-patient lipid profiles comparing the performance of Friedewald, Martin-Hopkins and the Sampson formulae to measure LDL-C assays. They found that the Martin-Hopkins formula best estimated calculated LDL-C at low LDL-C of ≤1.8mmol/L, and at moderate hypertriglyceridemia 1.7–4.5 mmol/L compared to other formulae [22]. In addition, a study by Martin et al. demonstrated that the Martin-Hopkins formula was more accurate than the Friedewald formula in low LDL-C concentrations [23]. With the current debate of measuring fasted vs non-fasted LDL-C samples, Sathiakumar et al. evaluated the accuracy of Martin-Hopkins and the Friedewald formulae in relation to fasting status [24]. Their findings were that the Martin-Hopkins formula outperformed the Friedewald formula in both fasted and non-fasted samples. Furthermore, the Martin-Hopkins formula was found to be superior at low LDL-C concentration particularly in non-fasted samples.

It is agreed that accurate measurement of LDL-C is pivotal in ensuring correct assessment of patients' cardiovascular risk and treatment of dyslipidemia, targeted at lowering their LDL concentration [25]. In 2021, ESC the published guidelines recommending that patients be first categorized according to the Systemic Coronary Risk Estimation 2 (SCORE2) and Systemic Coronary Risk Estimation 2-Older Persons (SCORE2-OP). The SCORE2 and SCORE2-OP estimates an individual's ten-year risk of fatal cardiovascular disease based on modifiable and non-modifiable risk factors. The categories include very-high, high, moderate, and low-risk [24]. Patient's with well controlled diabetes mellitus of less than 10 years' duration and no evidence of target organ damage are categorized as moderate risk. The high-risk category represents patients with diabetes mellitus without any atherosclerotic cardiovascular disease and with or without target organ damage. The very-high risk category represents patients with high-risk features and in addition have with renal impairment or the presence of microvascular disease [24]. Each category is assigned a specific LDL-C target concentration, emphasizing the importance of accurate and precise LDL-C methods to avoid misclassification and

(A)

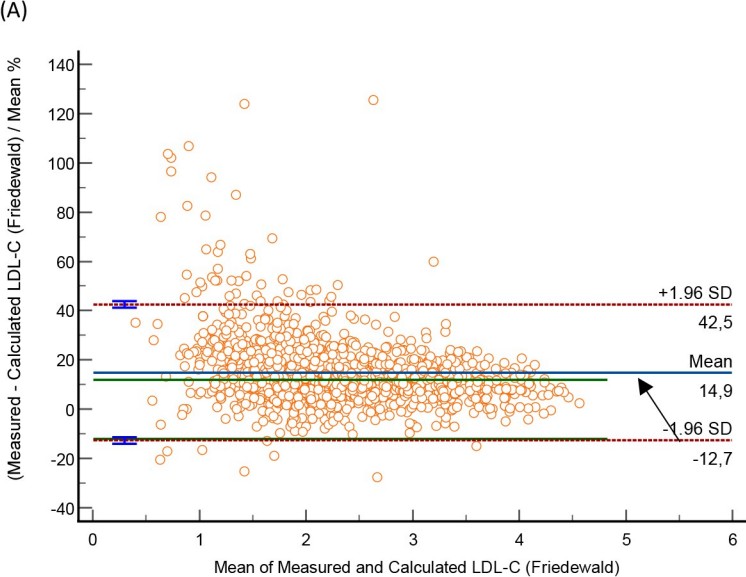

(B)

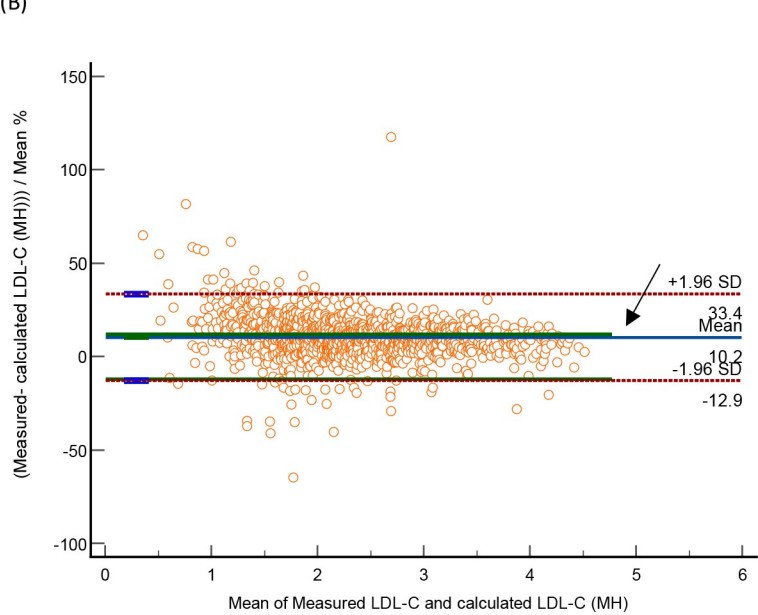

**Fig 2. Bland Altman Plots comparing measured LDL-C to calculated LDL-C.** Measured LDL-C to Friedewald formula (A) and measured LDL-C to Martin Hopkins formula (B).

mismanagement of patients. The higher an individual's SCORE2 and SCORE2-OP, the lower the target LDL-C. The SCORE2/SCORE2-OP for the very-high risk category is ≥10%, with an LDL-C treatment target of 1.4 mmol/L. The very-high risk category includes people with documented atherosclerotic cardiovascular disease (acute coronary syndrome, stable angina, coronary revascularization, stroke and transient ischemic attack), diabetes mellitus with target organ damage, type 1 diabetes mellitus for more than 20 years' duration, severe chronic kidney disease or familial hyperlipidemia with atherosclerotic cardiovascular disease [26].

**Table 3. The mean bias, regression equation, correlation coefficient and p-value of measured and calculated low-density lipoprotein at different triglycerides (TG) cut-offs.**

| | TG <1.7 mmol/L (N = 898) | | TG ≥1.7 mmol/L (N = 349) | |
|---|---|---|---|---|
| | **Friedewald Formula** | **Martin-Hopkins Formula** | **Friedewald Formula** | **Martin-Hopkins Formula** |
| Mean bias (95% Confidence interval) | 12.3% (11.5–13) | 12.6% (11.9–13.3) | 21.8% (19.9–23.8) | 4.2% (3.0–5.5) |
| Regression equation | y = -0.161 + 0.965 x | y = -0.148 + 0.957 x | y = -0.448 + 1.012 x | y = 0.071 + 0.920 x |
| Correlation coefficient | 0.963 | 0.962 | 0.949 | 0.949 |
| p-Value | <0.0001 | <0.0001 | <0.0001 | <0.0001 |

The Friedewald formula, which has long been used to calculate LDL-C levels, uses a fixed TG: cholesterol ratio as a proxy of VLDL-C, without considering chylomicrons which have higher TG than VLDL-C [12]. This is unlikely to be true in clinical practice, since patients are not always fasted. Therefore, the Friedewald formula is likely to overestimate VLDL-C and LDL-C in hypertriglyceridemia states and thus has been shown to be increasingly inaccurate at TG concentration between 2.3 to 4.5mmol/L [12]. Multiple studies have demonstrated that this formula underestimates LDL-C, resulting in cardiovascular risk misclassification [27,28]. As a result, numerous LDL-C formulae have been created, showing the benefits of being cost-effective, with a faster turnaround time, and simple compared to measured. There are, however, not without drawbacks. Several researchers have compared these LDL-C formulae in the past. Karkhaneh et al. [13] for example, compared eight equations in 2752 participants and found that the Friedewald formula overestimates LDL-C at TG values of 3.38mmol/L. A study by Sampson et al. also supported the evidence that the Martin-Hopkins formula was superior to the Friedewald formula in patients with low LDL-C as well as those with hypertriglyceridemia [24].

The Martin-Hopkins formula was first derived and validated in 2013 by Martin et al. in a study population of approximately 1.3 million fasted and nonfasted participants [16]. The Martin-Hopkins formula replaced the fixed TG: VLDL ratio with an adjustable factor derived from a 180 cell strata table by dividing TG by non-HDL. Non-HDL is calculated by subtracting HDL from total cholesterol. Martin et al. found that the Martin-Hopkins formula had good correlation to directly measured LDL-C compared to the Friedewald formula and further studies demonstrated this finding as well [16]. Our study has shown that the Martin-Hopkins formula is more accurate than Friedewald formula in diabetic patients. Therefore, it may be beneficial to apply the Martin-Hopkins formula when the measurement LDL-C concentration is not possible and inaccurate.

A similar conclusion was reached by Kang et al. when they evaluated four alternative LDL-C formulae in a Korean population and found that the Martin-Hopkins formula produced the least overestimation and underestimation of LDL-C values when compared to the Friedewald formula [15]. However, Lee et al. discovered that the Martin-Hopkins formula overestimated LDL-C in the Korean population, implying a possible racial variance [19], and underlining the need for additional validation of such formula in other populations.

Additional evidence comes from a study by Ferrinho et al. which found that while both the Friedewald and Martin-Hopkins formulae showed good correlation with measured LDL-C, the latter performed better in samples with low LDL-C (2.6 mmol/L) and diabetic participants [29]. Even though the Martin-Hopkins and Friedewald formulae all exhibited a good correlation to measured LDL-C, Song et al. found that the Martin-Hopkins formula was superior to the Friedewald formula across all TG values, especially in patients with dyslipidemias [30], the findings which are consistent with this study. These findings were confirmed by Sathiyakumar

(A)

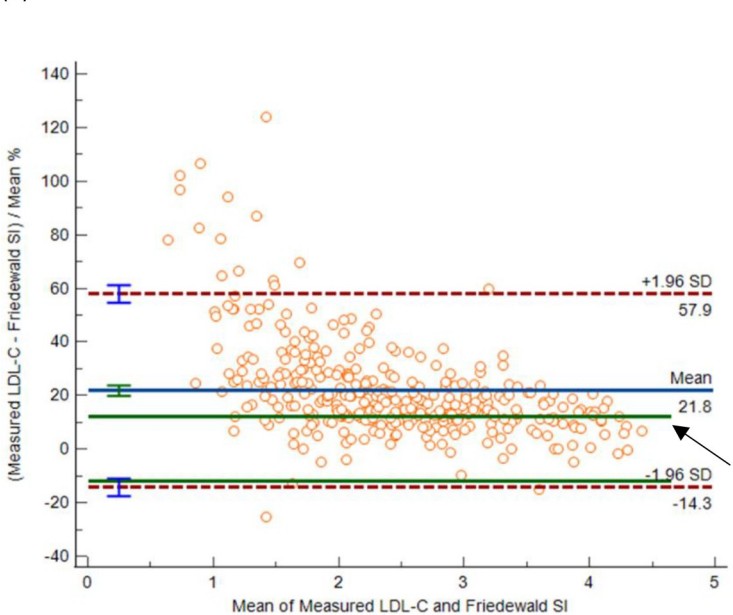

(B)

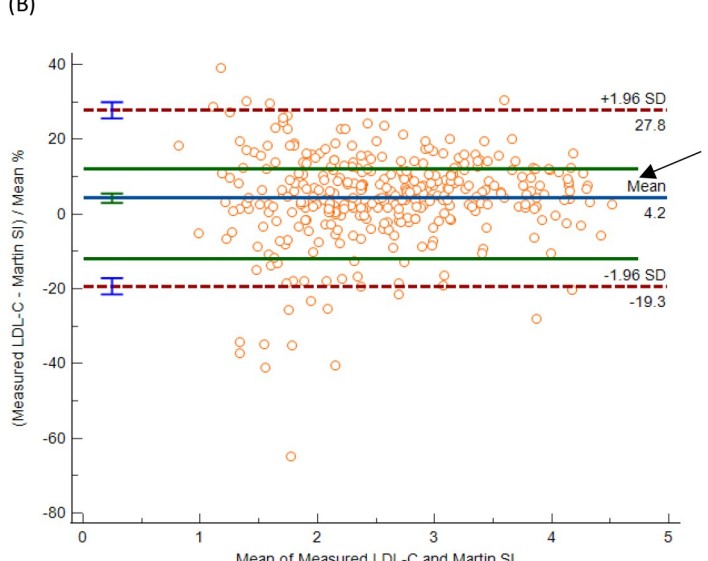

**Fig 3. Bland Altman Plot comparing measured LDL-C to the Friedewald formula and Martin Hopkins formula at TG-C of $\geq$ 1.7 mmol/L.** (A) There is a positive mean bias of 21.8% (95% confidence interval 19.9–23) between measured LDL-C and Friedewald formula. and a reduced positive bias of 4.2% (95% CI 3.0–5.5) with the (B) Martin-Hopkins formula. The total allowable limit of 12% is indicated by the arrow.

et al. who found that at TG concentrations of >4.5mmol/L, the Martin-Hopkins formula had a mean absolute deviation difference of 0.3, while the Friedewald formula had the lowest accuracy [31]. Jagesh et al. also demonstrated that the Martin-Hopkins formula achieved a better precision at higher TG concentrations compared to the Friedewald formula [32].

Miller et al. first demonstrated that the Roche Diagnostic method compared to the gold standard method had an imprecision ranging from 1.3–1.9% and a mean bias of -3.9% which is

within the recommended imprecision of $\leq 4\%$ and mean bias of $\leq 4\%$ by the NCEP ATP III as well an acceptable total error of 11% [33]. Furthermore, Miller et al. in 2010 showed that the direct Roche Diagnostic method had a persistent negative bias when compared to the gold standard method and this finding was more profound in patients that were treated for CVD [34].

## Strengths and limitations

Our study had several strengths: (i) Our data was derived from those who are most likely to develop dyslipidemia in a real-world environment and, (ii) it compared measured LDL-C to Friedewald and Martin-Hopkins across a wide range of TG concentrations. Yet, it may have shown some limitations due to the lack of comparison between the Roche assays and the reference method. The used Roche assay, which has its own bias compared to the reference method, cannot be assumed to represent "true LDL-C". However, homogeneous assays for measuring LDL-C are routinely used and recommended by the NCEP [35] and their performance is within specifications by NCEP ATP III [32]. Also, our database was not linked to clinical data, therefore, we were unable to assess the effect of medical conditions or medications on lipid profile effect, as this was a retrospective study.

We conclude that both the Martin-Hopkins and Friedewald equations have a strong correlation to measured LDL-C levels in the general population. The Martin-Hopkins formula outperformed the Friedewald formula in the South African diabetic population across all LDL concentrations, particularly at low concentrations of 1.4mmol/L and at hypertriglyceridemia of 1.7mmol/L. We demonstrated that the Martin-Hopkins formula generates more accurate findings than the Friedewald formula.

## Supporting information

**S1 Fig. Passing-Bablok plots at different low-density lipoprotein (LDL-C) treatment target concentrations using the Friedewald as well as the Martin-Hopkins formulae.** (A-B) LDL-C of <1.4 mmol/L. (C-D) LDL-C of 1.4–1.7 mmol/L. (E-F) LDL-C of 1.8–2.5 mmol/L. (G-H) LDL-C of 2.6–2.9 mmol/L and (J-K) LDL-C $\geq$3.0 mmol/L. The plots show the regression line (Solid blue line) and the confidence interval for the regression line (dashed lines). (DOCX)

**S1 Dataset.**
(XLSX)

## Acknowledgments

I would like to show my sincere gratitude to my supervisors Dr S Khoza and Dr N Kone along with Professor Jaya George the chemical Pathology head of department at the University of the Witwatersrand for the substantial contribution to this project.

## Author Contributions

**Conceptualization:** Mogomotsi Dintshi, Ngalulawa Kone, Siyabonga Khoza.

**Formal analysis:** Mogomotsi Dintshi.

**Supervision:** Ngalulawa Kone, Siyabonga Khoza.

**Writing – original draft:** Mogomotsi Dintshi.

**Writing – review & editing:** Ngalulawa Kone, Siyabonga Khoza.

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
