## [Decision Letter · Decision Letter 0]

21 Sep 2022

PONE-D-22-19758Comparison of measured LDL cholesterol with calculated LDL-cholesterol using the Friedewald and Martin-Hopkins formulae in diabetic adults at Charlotte Maxeke Johannesburg Academic Hospital/NHLS Laboratory.PLOS ONE

Dear Dr.Dintshi, 

Thank you for submitting your manuscript to PLOS ONE. After careful consideration, we feel that it has merit but does not fully meet PLOS ONE’s publication criteria as it currently stands. Therefore, we invite you to submit a revised version of the manuscript that addresses the points raised by the reviewer below.

We look forward to receiving your revised manuscript.

Kind regards,

Shukri AlSaif

Academic Editor

PLOS ONE

Journal Requirements:

"The authors state no conflict of interest."

4. PLOS requires an ORCID iD for the corresponding author in Editorial Manager on papers submitted after December 6th, 2016. Please ensure that you have an ORCID iD and that it is validated in Editorial Manager. To do this, go to ‘Update my Information’ (in the upper left-hand corner of the main menu), and click on the Fetch/Validate link next to the ORCID field. This will take you to the ORCID site and allow you to create a new iD or authenticate a pre-existing iD in Editorial Manager. Please see the following video for instructions on linking an ORCID iD to your Editorial Manager account: https://www.youtube.com/watch?v=_xcclfuvtxQ.

Reviewers' comments:

Reviewer's Responses to Questions

**Comments to the Author**

1. Is the manuscript technically sound, and do the data support the conclusions?

Reviewer #1: Yes

2. Has the statistical analysis been performed appropriately and rigorously? 

Reviewer #1: Yes

3. Have the authors made all data underlying the findings in their manuscript fully available?

Reviewer #1: No

4. Is the manuscript presented in an intelligible fashion and written in standard English?

Reviewer #1: Yes

5. Review Comments to the Author

Reviewer #1: This is a well written manuscript which seek to compare measured LDL-C with calculated LDL-C using the Friedewald and

Martin-Hopkins formulae in diabetic patients over a range of LDL-C and TG concentrations. The limitations of the Friedewald equation is well documented, subsequently, many studies has evaluated other formulae and compared with the Friedewald equation.

the current manuscript is well written, I however have the following questions or remarks;

did authors consider the medication of study participants?

did authors consider the disease duration? these factors are confounding and authors should answer and address their consequence on their lipid profile

Figure legends/titles should be below the figures and not above it, authors should revise as such.

6. PLOS authors have the option to publish the peer review history of their article (what does this mean?). If published, this will include your full peer review and any attached files.

Reviewer #1: No

---

## [Author Response · Author response to Decision Letter 0]

4 Nov 2022

Below is point-form response to the editor’s and reviewer’s comments.

• Editor Point P 0.1- Please ensure that your manuscript meets PLOS ONE's style requirements, including those for file naming.

Response: Thank you for providing us with the relevant templates for the requirements of PLOS ONE for research report. We have changed all major headings to font size 18 and also wrote them into sentence case. Figure captions changed as per PLOS ONE requirements, please see Results section page 9 to 14. Equations were formatted using the equation tool, see page 8. Table titles changed to bold. In-site citation changed to square brackets.

• Editor Point 0.2- Thank you for stating the following in your Competing Interests section: 

"The authors state no conflict of interest."

This information should be included in your cover letter; we will change the online submission form on your behalf

Response: Thank you for the above suggestion. We have included the “The authors declare that no competing interest exist” in the cover letter. We appreciate that you will change this on the online submission form on our behalf.

• Editor Point P0.3- We note that you have stated that you will provide repository information for your data at acceptance. Should your manuscript be accepted for publication, we will hold it until you provide the relevant accession numbers or DOIs necessary to access your data. If you wish to make changes to your Data Availability statement, please describe these changes in your cover letter and we will update your Data Availability statement to reflect the information you provide.

Response: We confirm that data mentioned in the manuscript will be made available and relevant accession numbers will be provided.

• Editor Point P 0.4- Please review your reference list to ensure that it is complete and correct. If you have cited papers that have been retracted, please include the rationale for doing so in the manuscript text, or remove these references and replace them with relevant current references. Any changes to the reference list should be mentioned in the rebuttal letter that accompanies your revised manuscript. If you need to cite a retracted article, indicate the article’s retracted status in the References list and also include a citation and full reference for the retraction notice.

Response: Thank you. No papers in the references have been retracted. 

Point-By-Point Reply to Reviewer’s Comments

• Reviewer Point P 1.1- Have the authors made all data underlying the findings in their manuscript fully available?

Reviewer #1: No

Response: Thank you very much for pointing it out. All raw data is uploaded on protocols.io

DOI: dx.doi.org/10.17504/protocols.io.yxmvm2ky6g3p/v1

(Private link for reviewers: https://www.protocols.io/private/37E720DD488211EDB54B0A58A9FEAC02 to be removed before publication.)

• Reviewer Point P 1.2- Review Comments to the Author

Reviewer #1: This is a well written manuscript which seek to compare measured LDL-C with calculated LDL-C using the Friedewald and

Martin-Hopkins formulae in diabetic patients over a range of LDL-C and TG concentrations. The limitations of the Friedewald equation is well documented, subsequently, many studies has evaluated other formulae and compared with the Friedewald equation.

the current manuscript is well written, I however have the following questions or remarks;

did authors consider the medication of study participants?

did authors consider the disease duration? these factors are confounding and authors should answer and address their consequence on their lipid profile

Figure legends/titles should be below the figures and not above it, authors should revise as such.

Response: Thank you for the positive feedback and highlighting the questions above. This was a retrospective study with limited access to participants’ medical records thus drug history and disease duration could not be obtained, therefore this was not assessed. We aimed to compare the analytical performance of measured LDL-C to calculated LDL-C over a wide range of LDL-C and triglyceride concentrations. We do acknowledge the effect of lipid lowering drugs on the lipid profile however they do not impact the analytical performance of LDL-C methods.

We have included our limited access to medical records as a limitation to our study and in addition recommended that further studies include participants’ drug history.

Figure titles and legends placement have been changed as suggested.

---

## [Editor Report · Decision Letter 1]

8 Nov 2022

Comparison of measured LDL cholesterol with calculated LDL-cholesterol using the Friedewald and Martin-Hopkins formulae in diabetic adults at Charlotte Maxeke Johannesburg Academic Hospital/NHLS Laboratory.

PONE-D-22-19758R1

Dear Dr. Dintshi,

We’re pleased to inform you that your manuscript has been judged scientifically suitable for publication and will be formally accepted for publication once it meets all outstanding technical requirements.

Kind regards,

Shukri AlSaif

Academic Editor

PLOS ONE
---

## [Editor Report · Acceptance letter]

5 Dec 2022

PONE-D-22-19758R1 

Comparison of measured LDL cholesterol with calculated LDL-cholesterol using the Friedewald and Martin-Hopkins formulae in diabetic adults at Charlotte Maxeke Johannesburg Academic Hospital/NHLS Laboratory. 

Dear Dr. Dintshi:

I'm pleased to inform you that your manuscript has been deemed suitable for publication in PLOS ONE. Congratulations! Your manuscript is now with our production department. 

Kind regards, 

on behalf of

Dr. Shukri AlSaif 

Academic Editor

PLOS ONE